# Informing Reinforcement Learning Agents by Grounding Natural Language to Markov Decision Processes

## Abstract

While significant efforts have been made to leverage natural language to accelerate reinforcement learning, utilizing diverse forms of language efficiently remains unsolved. Existing methods focus on mapping natural language to individual elements of MDPs such as reward functions or policies, but such approaches limit the scope of language they consider to make such mappings possible. We present an approach for leveraging general language advice by translating it to a grounded formal language capable of expressing information about *every* element of an MDP and its solution including policies, plans, reward functions, and transition functions. We also introduce a new model-based reinforcement learning algorithm called RLang-Dyna-Q that is capable of leveraging all such advice. We demonstrate the generality of our approach in a number of experiments by informing agents with a variety of natural language advice, leading to significant performance gains.

## 1 Introduction

Language serves as a powerful means for humans to share information about the world. Our grasp of language allows us to learn more quickly or even skip learning altogether, as we can learn to perform new tasks with ease by drawing upon the domain expertise of others in the form of advice. An open question in reinforcement learning is how language advice can be leveraged to speed up learning in Markov Decision Processes (MDPs), as learning to perform tasks *tabula rasa* is exceptionally difficult—and often impossible—in the real world. While many methods of leveraging advice for learning have emerged in the literature, a coherent theory of *language grounding* that can support the use of language for reinforcement learning has not.

Virtually all research in language and RL attempts to ground language to individual elements of MDPs such as policies (Liang et al., 2023; Vemprala et al., 2023; Wu et al., 2023; Andreas et al., 2017), reward functions (Squire et al., 2015), and goals (Colas et al., 2020). The main drawbacks of these works is that they restrict their approach to narrow fragments of natural language. For example, a statement like *"if a mug is tipped over, its contents will spill out"* clearly refers to a transition function, and mapping this information to a policy is not straightforward. For this reason, works that ground language to policies primarily focus on *imperative* sentences that naturally correspond to policies, plans, or reward functions. Likewise, works that ground language to transition functions focus mainly on *declarative* sentences, which may provide information about the dynamics of a domain. This consistent divergence in methodology suggests that not all language should be grounded to the same component of an MDP, and that a general language grounding system for reinforcement learning agents should be capable of grounding language to *every* element of an MDP and its solution.

We therefore propose a novel approach to grounding natural language for use in reinforcement learning that formulates the language grounding problem as a machine translation task from natural language to RLang (Rodriguez-Sanchez et al., 2023), a formal language designed to express information about every element of an MDP and its solution. Our approach is akin to semantic parsing (Mooney, 2007)—a problem in natural language understanding that involves translating natural language into a formal representation—as we think of RLang as a grounded formal language for MDPs that offers a systematic means of expressing knowledge about a task. Such an approach calls for a learning agent capable of leveraging all such MDP components, including a partial policy, reward function, plan,

and transition function. We therefore introduce RLang-Dyna-Q, a model-based tabular RL agent based on Dyna-Q Sutton et al. (1998), that can effectively leverage such advice. We demonstrate the strength and generality of our approach by grounding a variety of natural language advice to RLang programs, which RLang-Dyna-Q can use to significantly improve performance, sometimes making it possible to solve tasks that vanilla Dyna-Q cannot solve.

## 2 BACKGROUND

### 2.1 MODEL-BASED REINFORCEMENT LEARNING

Reinforcement learning tasks are typically modeled as Markov decision processes (MDPs), which can be represented by a tuple $\langle S, A, R, T, \gamma \rangle$, where $S$ is the set of states, $A$ is the set of actions, $R$ is the reward function, $T$ is the transition function, and $\gamma$ is the discount factor. The goal of an agent is to find a policy, $\pi(a|s)$—a function that selects an action for each state—which maximizes the expected sum of discounted rewards:

$$\mathbb{E}_\pi \left[ \sum_{t=0}^{\infty} \gamma^t R(s_t, a_t, s_{t+1}) \right].$$

Value-based reinforcement learning algorithms rely on estimating the optimal action-value function $q_*$, defined as

$$q_*(s, a) = \max_\pi q_\pi(s, a),$$

providing the expected return for taking action $a$ in state $s$ and subsequently following an optimal policy (Sutton et al., 1998). Q-learning (Watkins, 1989) works to approximate $q_*$ by applying the following update rule after taking roll-outs in the environment:

$$Q(S_t, A_t) \leftarrow Q(S_t, A_t) + \alpha \left[ R_{t+1} + \gamma \max_a Q(S_{t+1}, a) - Q(S_t, A_t) \right].$$

Building on Q-learning, Dyna-Q (Sutton et al., 1998) introduces an additional component: a model of the environment. While Q-learning learns from direct interaction with the environment alone, Dyna-Q builds an internal model of the environment and updates the action-value function using both real and simulated roll-outs, enabling faster convergence to the optimal action-value function.

### 2.2 LARGE LANGUAGE MODELS FOR MACHINE TRANSLATION

Large Language Models (LLMs), often based on architectures like the Transformer (Vaswani et al., 2017), are trained to predict the next token $x_t$ in a sequence given the preceding tokens $\{x_1, x_2, ..., x_{t-1}\}$ via the following objective:

$$\mathcal{L} = -\sum_t \log P(x_t | x_1, x_2, \ldots, x_{t-1}).$$

In very large models, this objective results in emergent capabilities such as natural language understanding and generation, making them suitable for a variety of tasks beyond mere text completion including question-answering, summarization, and more (Bubeck et al., 2023). One useful emergent capability of LLMs is machine translation, the translation of text from one language to another. While specialized neural machine translation systems are trained using a parallel corpus to maximize the conditional probability $P(y|x)$, where $x$ is the source sequence and $y$ is the target sequence (Bahdanau et al., 2014), LLMs have achieved similar translation capabilities despite not being trained explicitly on this objective (Brown et al., 2020). Furthermore LLMs have been shown to be proficient at generating text in formal languages such as Python given a natural language prompt (Chen et al., 2021; Li et al., 2023).

### 2.3 LEVERAGING FORMAL SPECIFICATION LANGUAGES FOR DECISION-MAKING

Formal specification languages have long been a useful tool to inform decision-making agents. In the field of classical planning, for example, it is standard to use the Planning Domain Description

Table 1: Selected MDP elements, corresponding RLang groundings, and natural language interpretations. The first column shows a component of the MDP, the second shows an RLang expression that can inform it, and the last column contains a description of the expression.

| MDP Component | RLang Declaration | Natural Language Interpretation |
|---|---|---|
| Policy
$\pi : \mathcal{S} \times \mathcal{A} \rightarrow [0, 1]$ | ```Policy build_bridge:```
```    if at_workbench:```
```        Execute use```
```    else:```
```        Execute go_to(workbench)``` | If you are at a workbench, use it. Otherwise, go to it. |
| Plan
$\{A_0, A_1, ..., A_n\}$ | ```Plan gather_materials:```
```    Execute go_to(wood)```
```    Execute pickup```
```    Execute go_to(string)```
```    Execute pickup``` | Go to the wood and pick it up, then go to the string and pick it up. |
| Reward, Transition Func.
$R_e : \mathcal{S} \times \mathcal{A} \times \mathcal{S} \rightarrow \mathbb{R}$
$T_e : \mathcal{S} \times \mathcal{A} \times \mathcal{S} \rightarrow [0, 1]$ | ```Effect common_sense:```
```    if at(Wall) and A == walk:```
```        Reward 0```
```        S' -> S```
```    if at(Lava) and A == walk:```
```        Reward -1```
```        S' -> S*0``` | Walking into walls will get you nowhere. Walking into lava will kill you. |

Language (PDDL; Ghallab et al. 1998) and its probabilistic extension PPDDL (probabilistic PDDL; Younes & Littman, 2004) to specify the complete dynamics of an environment. Other languages like Linear Temporal Logic (LTL; Littman et al., 2017; Jothimurugan et al., 2019) and Policy Sketches (Andreas et al., 2017) are sufficient for describing goals and hierarchical policies, respectively, for instruction-following agents. While effective, one limiting factor of these formal languages is their narrow scope. Natural language, by contrast, can be used to express information about nearly *all* the elements of decision-making.

A recent formal language to emerge from the literature is RLang (Rodriguez-Sanchez et al., 2023). While previous formal languages for decision-making narrowly focus on individual components of an MDP such as a policy or reward function, RLang was designed to provide information about *every* component of a structured Markov Decision Process and its solution. Formally, an RLang specification is a set of RLang groundings $\mathcal{G}$ given by an RLang program $\mathcal{P}$ and an RLang vocabulary $\mathcal{V}$, which may include additional groundings for use across multiple MDPs. Some example RLang programs and their natural language interpretations can be seen in Table 1. Crucially, advice specified by RLang can be compiled directly into many components of an MDP—including policies, transition functions, reward functions, and plans—for direct integration into existing RL agents.

## 3 GROUNDING NATURAL LANGUAGE TO MDPS

One major motivation for leveraging language advice in reinforcement learning is to supply agents with the kinds of commonsense reasoning that language can easily express. Consider the LavaCrossing environment in Figure 1. Any human playing the game would quickly learn that walking into the lava squares kills you, or likewise that walking into walls will do nothing at all. For humans, communicating this knowledge to others with language is natural, but leveraging such language advice in reinforcement learning is a major unsolved problem. An alternative approach to supplying common sense advice to RL agents involves specifying it in a formal language relevant to decision-processes, which can more straightforwardly be used by a learning agent to improve learning. One such formal language to emerge from the literature is RLang (Rodriguez-Sanchez et al., 2023), a highly expressive language that was designed to capture information about *every* element of an MDP and its solution.

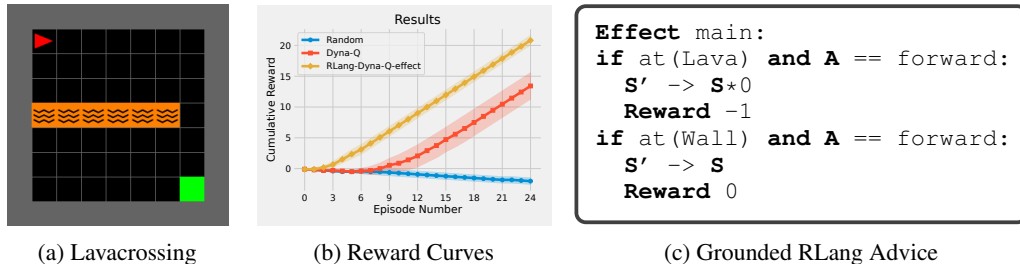

(a) Lavacrossing     (b) Reward Curves     (c) Grounded RLang Advice

Figure 1: **LavaCrossing Experiment.** The agent was given the following advice: "Walking into lava will kill you. Walking into walls will do nothing."

### 3.1 GROUNDING NATURAL LANGUAGE ADVICE TO RLANG PROGRAMS

We hypothesize that RLang is a sufficient representation for natural language advice in the context of reinforcement learning, and formulate the language grounding problem as a machine translation task from natural language to RLang. Our task is as follows: given an RLang vocabulary $\mathcal{V}$ for a given MDP and a piece of natural language advice $u$, we seek a function $\phi : u \times \mathcal{V} \to \mathcal{P}_u$, where $\mathcal{P}_u$ is an executable RLang program capturing the advice in $u$.

We propose to do this translation using a pre-trained large language model in a two-stage pipeline by 1) identifying which RLang grounding type would best capture the language advice; and 2) few-shot translating the advice into an RLang program. Stage 1, the selection stage, instructs the LLM to classify a novel piece of advice $u$ into RLang grounding types such as Effects, Policies, and Plans, consulting a small number of example classifications in the prompt. We assume that each piece of advice—which may contain multiple sentences—grounds to a single RLang grounding type. Stage 2, the translation stage, instructs the LLM to translate $u$ to an RLang program specifying the grounding type given by Stage 1 using roughly 5 example translations in the prompt that were hand-engineered to cover a wide range of RLang's syntax. This pipeline effectively grounds the advice, yielding an RLang program that may contain partial transition functions, reward functions, policies, and plans. Our pipeline is illustrated in Figure 2.

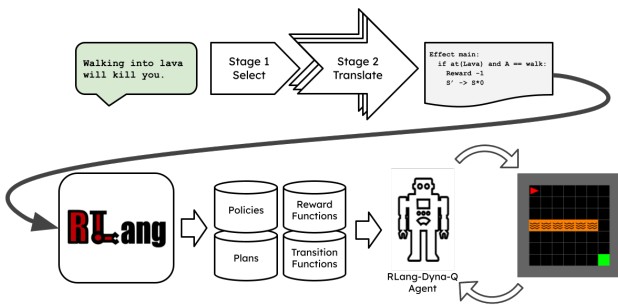

Figure 2: Our pipeline for translating natural language advice to RLang. We extend the existing RLang pipeline to include natural language translation and a Dyna-Q agent capable of leveraging all forms of RLang advice.

### 3.2 RLANG-DYNA-Q: A SINGLE AGENT FOR LEVERAGING ALL OF RLANG

In the original RLang paper, the authors presented a number of RLang-enabled agents—including ones based on Q-Learning, PPO (Schulman et al., 2017), and DOORmax (Diuk et al., 2008)—each capable of leveraging *individual* RLang groundings to improve learning. However, leveraging general language advice requires integrating potentially *all* RLang groundings into a single learning agent. We therefore introduce RLang-Dyna-Q, a learning agent based on Dyna-Q (Sutton et al., 1998) that is capable of simultaneously leveraging a partial policy, plan, reward function, and transition function given by an RLang program. Dyna-Q is an appropriate core learning agent because integrating

policies and dynamics is most natural in a model-based learning algorithm (see Algorithm 1, our modifications to Dyna-Q are in blue).

---

**Algorithm 1** RLang-Dyna-Q Agent

---

**Given:** $\pi_{\text{RLang}}$, $T_{\text{RLang}}$, $R_{\text{RLang}}$ from an RLang program
Initialize $Q(s,a)$, $T(s,a)$, $R(s,a)$ for all $s \in \mathcal{S}$, $a \in \mathcal{A}(s)$
**loop**
    $s \leftarrow$ current (nonterminal) state
    With some prob. $\epsilon_2$, we execute the plan or policy given by RLang, in that order
    $a \leftarrow \epsilon_1, \epsilon_2\text{-greedy}(s, \pi_{\text{RLang}}, Q)$
    Execute action $a$; observe resultant state $s'$, and reward $r$
    $Q(s,a) \leftarrow Q(s,a) + \alpha\left[r + \gamma \max_{a'} Q(s',a') - Q(s,a)\right]$
    $T(s,a), R(s,a) \leftarrow s', r$       (assuming deterministic environment)
    **for** $i = 1$ to $N_1$ **do**
        $s \leftarrow$ random previously observed state
        $a \leftarrow$ random action previously taken in $s$
        $s', r \leftarrow T(s,a), R(s,a)$
        $Q(s,a) \leftarrow Q(s,a) + \alpha\left[r + \gamma \max_{a'} Q(s',a') - Q(s,a)\right]$
    **end for**
    **for** $i = 1$ to $N_2$ **do**
        $s \leftarrow$ random previously observed state
        $a \leftarrow$ random action **not** previously taken in $s$
        Predict $s', r$ using dynamics given by RLang
        $s', r \leftarrow T_{\text{RLang}}(s,a), R_{\text{RLang}}(s,a)$
        $Q(s,a) \leftarrow Q(s,a) + \alpha\left[r + \gamma \max_{a'} Q(s',a') - Q(s,a)\right]$
    **end for**
**end loop**

---

## 4 EXPERIMENTS

We set forth two hypotheses: 1) RLang is an expressive grounding for natural language advice in the context of RL; and 2) we can leverage natural language advice to improve the performance of a learning agent by grounding natural language to RLang. To evaluate these hypotheses we ran two sets of experiments: one for assessing the expressive power of RLang as a grounding for natural language and another for assessing whether our language grounding pipeline can effectively improve performance on an RL task.

In both sets of experiments we utilize custom environments from the Minigrid/BabyAI library (Chevalier-Boisvert et al., 2023; 2018), a platform for studying the behavior of language-informed agents. In a typical Minigrid environment, an agent might reason about opening and closing doors using keys which may be hidden in other rooms, managing a small inventory of items, removing obstacles like balls out of the way to reach other rooms or objects, and avoiding lava, all for the ultimate purpose of reaching a goal. Minigrid environments are an ideal setting for our experiments for three reasons: 1) they can be solved using tabular RL algorithms, which our informed, model-based RLang-Dyna-Q agent is based on; 2) there are clear and obvious referents of language in both the state and action spaces of these environments (e.g. keys, doors, and balls are represented neatly in a discrete state space and skills such as walking towards objects are easy to implement); 3) many objects are shared across environments enabling the reuse of a common RLang vocabulary for referencing these objects, which makes it easier for our translation pipeline to ground novel advice.

For each environment, we provide a set of RLang groundings to the agent that act as targets for natural language translation. These groundings include perception abstractions such as the objects in the environment (e.g., `yellow_key`, `red_door`) and a short list of predicates for reasoning with them (e.g., `carrying()`, `reachable()`, `at()`), as well as a single abstract action in the form of a lifted skill for walking to any reachable object (`go_to()`). All agents we run experiments on, including the Random, Dyna-Q, and RLang-Dyna-Q agents, are given access to this lifted skill. However, we do not provide the Dyna-Q agent with any perception abstractions, as including them induces an equivalent state space in the tabular RL setting. Likewise, the Random agent does not

Table 2: **User Study.** We collected 10 pieces of advice from 10 undergraduate students for the LockedRoom environment. For each piece of advice, 5 agent instances were run for 25 episodes on the LockedRoom environment for 500 steps. The cumulative discounted reward for the 25 episodes is in the first column along with a 95% confidence interval. The average percent increase in cumulative discounted reward over the baseline is present in the second column. The second-to-last piece of advice did not ground to a valid RLang program, so no experiment was run.

| Avg Cumulative Return | % improvement | Natural Language Advice |
|---|---|---|
| $17.86 \pm 2.36$ | — | No advice |
| $22.01 \pm 0.71$ | $+23.24$ | "Remember to toggle to open doors." |
| $16.79 \pm 1.75$ | $-5.99$ | "You don't need to carry keys to open the grey door." |
| $17.22 \pm 1.75$ | $-3.60$ | "Identify the room with the red key, move to that room by opening the door. Pick up the key. Identify the room with the red door, proceed there. Open the red door. Find the green square and go there to finish the game." |
| $23.55 \pm 0.69$ | $+31.86$ | "Move to the grey door, open it and enter the room until you get to the red key, pick it up. Exit the room and move towards the red door, open it and get into that room. Move to the green block and enter it." |
| $23.93 \pm 0.37$ | $+33.99$ | "Go to the grey door. open the grey door. go to the red key. pick up the red key. go to the red door. open the red door. go to the green square." |
| $24.07 \pm 0.22$ | $+34.77$ | "Pick up the red key after opening the grey door. Then walk to the red door, open it, and go to the goal." |
| $17.57 \pm 0.78$ | $-1.63$ | "You cannot open the red door without a red key." |
| $17.77 \pm 0.54$ | $-0.50$ | "Walking towards the red door is not very useful if it is closed." |
| — | — | "Go down until the second door on the left and pick up the key. Then exit the room and go down until the next door on the left and use it to open the door and get to the green box." |
| $18.35 \pm 1.93$ | $+2.76$ | "Go to the room that has the red key, pick it up, and then go to the room with a red door. Enter the room, and go to the green goal object." |

consider the state when selecting an action. In Stage 2 of the translation pipeline, we provide the list of available RLang groundings that can be referenced in an RLang program along with the language advice. This prevents the LLM from hallucinating imaginary skills, objects, and predicates when translating the advice into an RLang program. The LLM never interacts with the MDP directly. The translation examples used in the prompts in both stages of translation did not change across experiments, though these translations are vocabulary-specific and grounding advice to environments outside of Minigrid will require domain-compatible example translations.

## 4.1 EVALUATING TRANSLATION PIPELINE AND RLANG EFFICACY

To assess RLang's ability to capture the breadth of general language advice, we ran a small user study. After explaining the controls for navigation, opening and closing doors, and picking up and dropping objects, we asked 10 undergraduate students to play the LockedRoom MiniGrid task (pictured in Figure 3) until they felt they were proficient at it. Afterwards, we asked them to describe in one or two sentences any advice they would give to an agent completing the task for the first time. We collected their responses and ran them through our translation pipeline to arrive at the RLang groundings in Table 3 of the Appendix. Of 10 pieces of advice collected, 9 were translated into valid RLang programs, while 1 referenced groundings that did not exist (e.g. `second_left_door`). We used the remaining valid RLang programs to inform 9 separate RLang-Dyna-Q agents that we compared against a baseline Dyna-Q agent given no advice. With a few exceptions, providing advice either did not meaningfully impact performance over the baseline or led to dramatic improvements in performance (see Table 2). In the cases where advice did not impact performance, advice was translated into a valid RLang program, but the program was missing a crucial step because it did not have an appropriate RLang grounding to reference (e.g. it had no way of translating "the room with the red key"). This is not a limitation of our pipeline, however, as a more expressive RLang vocabulary file could be used to achieve such a translation (e.g. `room_with(red_key)` can evaluate to `grey_door`).

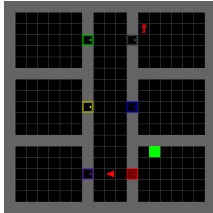

Figure 3: The initial state of the LockedRoom environment.

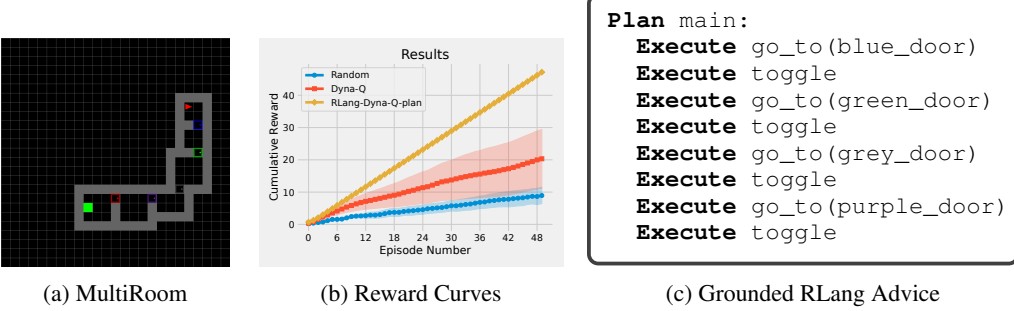

| (a) MultiRoom | (b) Reward Curves | (c) Grounded RLang Advice |

Figure 4: **MultiRoom Experiment.** The agent was given the following advice: "First go to the blue door, then the green door, then the grey door, then the purple door."

## 4.2 EVALUATING AGENT PERFORMANCE GIVEN EXPERT ADVICE

We evaluated the strength of our grounding approach on four diverse Minigrid environments—LavaCrossing, MultiRoom, MidMazeLava, and HardMaze—with the final two being custom environments that we engineered to make significantly more difficult for a tabular RL agent. For each environment, we collected multiple pieces of natural language advice from human experts and translated them into RLang programs using our two-stage pipeline. Each piece of advice was translated to a single RLang grounding type, and each piece of advice contained multiple sentences. We then ran our RLang-Dyna-Q agent on the environment with the translated RLang program. In one environment where multiple pieces of advice were given that grounded to different RLang types (e.g., both plans and effects), we performed an ablation study by running separate agents utilizing only one type of advice at a time—Effects, Plans, or Policies—to isolate the impact of each on performance. This allowed us to empirically validate the benefit of a combined RLang-Dyna-Q agent that leverages multiple types of advice simultaneously.

In all of the experiments, all forms of RLang-Dyna-Q significantly outperformed vanilla Dyna-Q. In LavaCrossing (see Figure 1), the agent is tasked with reaching a goal while avoiding lava, and merely advising the agent about the dangers of lava and uselessness of walking into walls greatly increases performance. In the MultiRoom environment (see Figure 4), in which the agent must open a series of doors to reach a goal, providing a plan in natural language significantly increased performance. In MidMazeLava (see Figure 5) and HardMaze (see Figure 6), the agent is faced with significantly more difficult tasks. In the former, the agent must unblock doors and open them with keys to reach a goal while avoiding lava, and in the latter the agent must traverse through many rooms, bringing keys across rooms to doors which must be unblocked to reach a goal. We collected paragraphs-worth of advice for these environments, which we translated into RLang plans, policies, and effects. In HardMaze, this language advice made it possible to solve the task, as the vanilla Dyna-Q agent did no better than random.

In all experiments, 10 instances of each agent were run per experimental condition to generate a 95% confidence interval on their cumulative reward over 50 episodes (LavaCrossing was run for 25 episodes only). The number of timesteps per episode varied across environments.

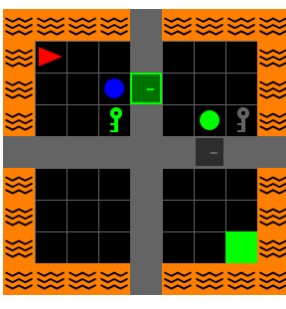

(a) Initial State

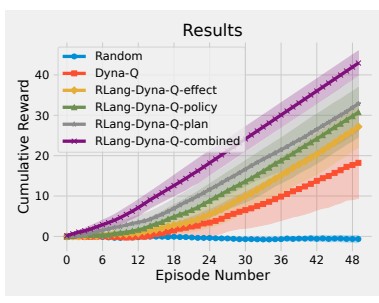

(b) Reward Curves

"Pick up the blue ball and drop it to your right. Then pick up the green key and unlock the green door. Then drop the key to your right." "Some general advice: If you are carrying a key and its corresponding door is closed, open the door if you are at it, otherwise go to the door if you can reach it. Otherwise, drop any keys for doors you can't reach. If you can reach the goal, go to it." "Walking into lava will kill you. If you're not at a door, toggling will do nothing. Trying to pick something up while you're carrying something is pointless. Walking into walls will do nothing."

(c) Language Advice

Figure 5: **MidMazeLava Experiment.** Language advice given to the agent was grounded to RLang effects, plans, and policies. The full translated RLang program is available in the appendix. All RLang-Dyna-Q agents outperformed Dyna-Q on this task.



(a) HardMaze Initial State

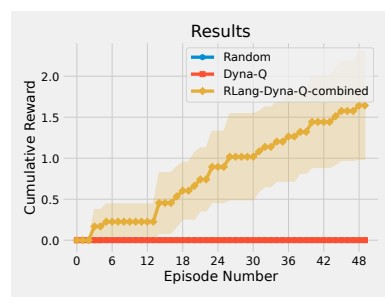

(b) Reward Curves

"Go and pick up the green ball, and drop it on your left, and then go pick up the blue key, and go to the blue door and open it up and drop the key on your left, and then go pick up the green key, and go to the green door to open it and drop the key on your left, and then go pick up the purple ball and drop it on your right." "Nothing will happen if you walk towards the wall, or try to open a purple door without the purple key if it is locked. The applies for the yellow door and key as well as the red door and key." "If you can reach the grey door and it is closed but you have the key, open it if you are at it or otherwise go to it. The same applies to the purple door, yellow door, and red door. Lastly, if you find the goal is reachable just go to the goal directly."

(c) Language Advice

Figure 6: **HardMaze Experiment.** Language advice given to the agent was grounded to RLang effects, plans, and policies. The full translated RLang program is available in the appendix. Vanilla Dyna-Q was not able to complete this task.

## 5 RELATED WORKS

**Language Use in Reinforcement Learning** Luketina et al. (2019) identify two variations of language usage in the reinforcement learning literature. The first variation, **language-conditional** RL, is one in which language use is a necessary component of the task. This includes environments where agents must execute commands in natural language (Mirchandani et al., 2021), or otherwise deal with language that is part of the MDP, e.g., in the observation or action space (Fulda et al., 2017; Kostka et al., 2017). The second variation is **language-assisted** RL, in which natural language is used to communicate task-related information to an agent that is *not necessary* for solving the task. In these settings, language can be used to inform policy structure (Watkins et al., 2023), reward functions (Goyal et al., 2019), transition dynamics (Narasimhan et al., 2018), or Q-functions (Branavan et al., 2012).

**Grounding Natural to Formal Languages for Planning and Learning** The notion of grounding natural language to a formal language for use in learning and planning is not new. Gopalan et al. (2018) and Berg et al. (2020) translate natural language commands into Linear Temporal Logic (LTL), which they use as reward functions for a learning agent or planning objectives, and Silver et al. (2023) and Miglani & Yorke-Smith (2020) ground natural language into PDDL, which is fed to a recurrent neural network to output solution plans. However, the advancement of large language models (LLMs) has led to even more capable agents that for leveraging formal languages. In the planning literature, Ahn et al. (2022); Huang et al. (2022); Song et al. (2023) use primitive formal languages for executing policies on real robots or in embodied environments, Liu et al. (2023a); Xie et al. (2023) translate natural language commands into PDDL plans with the help of LLMs, and Liu et al. (2023b) proposed a modular system to ground natural language into LTL formulas. Code is also a popular choice for formal languages: Liang et al. (2023); Vemprala et al. (2023); Wu et al. (2023) use an LLM to generate Python functions as policies from natural language instructions; Singh et al. (2022) also generates programs by prompting LLMs for code completion. For learning, more recent works focus on reward design with LLMs for RL agents: Yu et al. (2023) specifies reward with LLMs through code generation. Du et al. (2023) leverage commonsense reasoning for designing reward functions.

## 6 DISCUSSION AND CONCLUSION

Natural language grounding (Steels & Hild, 2012) is a popular topic with critical implications for all of AI. Just as RL is intended as a model of the totality of intelligent decision making, we propose that its core formalisms offer a natural target for language grounding. If MDPs are a good model of human decision-making, and humans invented language to share information that aids their decision-making, then the appropriate target for language grounding should be an MDP, or a richer and perhaps more structured decision process reflecting the complexity of human decision-making. One line of evidence for this claim is the direct correspondence between parts of speech and elements of structured decision-processes (Patel et al., 2020). For example, the object classes in Object Oriented MDPs (Diuk et al., 2008) naturally correspond to the concept of **common nouns** requiring **determiners** to single out class instances, and the parameters in Parameterized Action MDPs Masson et al. (2016) naturally correspond to **adverbs** for modifying the execution of discrete macro-actions (**verbs**).

More practically, knowledge expressed in natural language has immense potential to inform reinforcement learning agents, and thereby alleviate the high sample complexity of having to learn *tabula rasa*. We present a novel method for leveraging general natural language advice to expedite learning in Markov Decision Processes by translating it into RLang, a formal language designed to specify information about every element of an MDP and its solution. Our method can ground advice to reward functions, transition functions, plans, and policies. We also introduce a modified Dyna-Q agent capable of leveraging all of the types of information present in the partial MDP specification output by RLang. Our findings show that our approach can leverage a wide variety of language advice to accelerate learning.

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

# A  APPENDIX

## A.1  PROMPTS USED FOR TRANSLATION PIPELINE

### A.1.1  PROMPT USED FOR STAGE 1 OF THE TRANSLATION PIPELINE. GIVEN A NEW PIECE OF ADVICE, WE PROMPT THE LLM TO CLASSIFY IT AS AN EFFECT, PLAN, OR POLICY.

RLang is a formal language for specifying information about every element of a Markov Decision Process (S,A,R,T). Each RLang object refers to one or more elements of an MDP. Here is a description of three important RLang groundings:

Policy:  a direct function from states to actions, best used for more general commands.
Effect: a prediction about the state of the world or the reward function.
Plan: a sequence of specific steps to take.

Your task is to decide which RLang grounding most naturally corresponds to a given piece of advice:
Advice = "Don't touch any mice unless you have gloves on."
Grounding: Effect
Advice = "Walking into lava will kill you."
Grounding: Effect
Advice = "First get the money, then go to the green square."
Grounding: Plan
Advice = "Go through the door to the goal."
Grounding: Plan
Advice = "If you have the key, go to the door, otherwise you need to get the key."
Grounding: Policy
Advice = "If there are any closed doors, open them."
Grounding: Policy

### A.1.2  PROMPT USED FOR STAGE 2 OF THE PIPELINE TO TRANSLATE A PIECE OF ADVICE INTO AN RLANG PLAN.

Your task is to translate natural language advice to RLang plan, which is a sequence of specific steps to take. For each instance, we provide a piece of advice in natural language, a list of allowed primitives, and you should complete the instance by filling the missing plan function. Don't use any primitive outside the provided primitive list corresponding to each instance, e.g., if there is no 'green_door' in the primitive list you must not use 'green_door' for the plan function.

Advice = "Open the door with the key and go through it to the goal"
Primitives = ['Agent', 'Wall', 'GoalTile', 'Lava', 'Key', 'Door', 'Box', 'Ball', 'left', 'right', 'forward', 'pickup', 'drop', 'toggle', 'done', 'pointing_right', 'pointing_down', 'pointing_left', 'pointing_up', 'go_to', 'step_towards', 'yellow_key', 'yellow_door', 'agent', 'goal', 'at', 'in_inventory']

```
Plan main:
    Execute go_to(yellow_key)
    Execute pickup
    Execute go_to(yellow_door)
    Execute toggle
    Execute go_to(goal)
```

Advice = "Get the key behind the red door to open the grey door. Then drop the key to the left."
Primitives = ['Agent', 'Wall', 'GoalTile', 'Lava', 'Key', 'Door', 'Box', 'Ball', 'left', 'right', 'forward', 'pickup', 'drop', 'toggle', 'done', 'pointing_right', 'pointing_down', 'pointing_left', 'pointing_up', 'go_to', 'step_towards', 'yellow_key', 'yellow_door', 'agent', 'goal', 'at', 'in_inventory']

```
Plan main:
    Execute go_to(red_door)
    Execute toggle
    Execute go_to(grey_key)
    Execute pickup
    Execute go_to(grey_door)
    Execute toggle
    Execute left
    Execute drop
```

A.1.3 Prompt used for Stage 2 of the pipeline to translate a piece of advice into an RLang policy.

Your task is to translate natural language advice to RLang policy, which is a direct function from states to actions. For each instance, we provide a piece of advice in natural language, a list of allowed primitives, and you should complete the instance by filling the missing policy function. Don't use any primitive outside the provided primitive list corresponding to each instance, e.g., if there is no 'green_door' in the primitive list you must not use "green_door' for the policy function.

Advice = "If the yellow door is open, go through it and walk to the goal. Otherwise open the yellow door if you have the key."
Primitives = ['Agent', 'Wall', 'GoalTile', 'Lava', 'Key', 'Door', 'Box', 'Ball', 'left', 'right', 'forward', 'pickup', 'drop', 'toggle', 'done', 'pointing_right', 'pointing_down', 'pointing_left', 'pointing_up', 'go_to', 'step_towards', 'yellow_key', 'yellow_door', 'agent', 'goal', 'at', 'carrying']

```
Policy main:
  if yellow_door.is_open:
    Execute go_to(goal)
  elif carrying(yellow_key) and at(yellow_door) and not yellow_door.
  is_open:
    Execute toggle
```

Advice = "If you don't have the key, go get it."
Primitives = ['Agent', 'Wall', 'GoalTile', 'Lava', 'Key', 'Door', 'Box', 'Ball', 'left', 'right', 'forward', 'pickup', 'drop', 'toggle', 'done', 'pointing_right', 'pointing_down', 'pointing_left', 'pointing_up', 'go_to', 'step_towards', 'grey_key', 'red_door', 'grey_door', 'agent', 'purple_ball', 'at', 'carrying']

```
Policy main:
  if at(grey_key):
    Execute pickup
  elif not carrying(grey_key):
    Execute go_to(grey_key)
```

Advice = "If you are carrying a ball and its corresponding box is closed, open the box if you are at it, otherwise go to the box if you can reach it."
Primitives = ['Agent', 'Wall', 'GoalTile', 'Lava', 'Key', 'Door', 'Box', 'Ball', 'left', 'right', 'forward', 'pickup', 'drop', 'toggle', 'done', 'pointing_right', 'pointing_down', 'pointing_left', 'pointing_up', 'go_to', 'step_towards', 'green_ball', 'green_box', 'purple_box', 'agent', 'purple_ball', 'at', 'reachable', 'carrying']

```
Policy main:
  if carrying(green_ball) and not green_box.is_open:
    if at(green_box):
      Execute toggle
    elif reachable(green_box):
      Execute go_to(green_box)
```

Advice = "Drop any balls for boxes you can't reach"
Primitives = ['Agent', 'Wall', 'GoalTile', 'Lava', 'Key', 'Door', 'Box', 'Ball', 'left', 'right', 'forward', 'pickup', 'drop', 'toggle', 'done', 'pointing_right', 'pointing_down', 'pointing_left', 'pointing_up', 'go_to', 'step_towards', 'green_ball', 'green_box', 'purple_box', 'agent', 'purple_ball', 'at', 'reachable', 'carrying']

```
Policy main:
  if carrying(green_ball) and not reachable(green_box):
    Execute drop
  if carrying(purple_ball) and not reachable(purple_box):
    Execute drop
```

Advice = "if you have any key for a door that you cannot reach, you should drop it"
Primitives = ['Agent', 'Wall', 'GoalTile', 'Lava', 'Key', 'Door', 'Box', 'Ball', 'left', 'right', 'forward', 'pickup', 'drop', 'toggle', 'done', 'pointing_right', 'pointing_down', 'pointing_left', 'pointing_up', 'go_to', 'step_towards', 'green_ball', 'green_box', 'purple_box', 'agent', 'purple_ball', 'at', 'reachable', 'carrying']

```
Policy main:
  if carrying(green_key) and not reachable(green_door):
    Execute drop
  if carrying(purple_key) and not reachable(purple_door):
    Execute drop
  if carrying(red_key) and not reachable(red_door):
    Execute drop
```

Advice = "Hey listen, you can open the door if you have the key and at the door when the door is closed"
Primitives = ['Agent', 'Wall', 'GoalTile', 'Lava', 'Key', 'Door', 'Box', 'Ball', 'left', 'right', 'forward', 'pickup', 'drop', 'toggle', 'done', 'pointing_right', 'pointing_down', 'pointing_left', 'pointing_up', 'go_to', 'step_towards', 'green_ball', 'green_box', 'purple_box', 'agent', 'purple_ball', 'at', 'reachable', 'carrying']

```
Policy main:
  if carrying(purple_key) and not purple_door.is_open and at(
  purple_door):
    Execute toggle
```

A.1.4  PROMPT USED FOR STAGE 2 OF THE PIPELINE TO TRANSLATE A PIECE OF ADVICE INTO
       AN RLANG EFFECT.

---

Your task is to translate natural language advice to RLang effect, which is a prediction about
the state of the world or the reward function. For each instance, we provide a piece of advice
in natural language, a list of allowed primitives, and you should complete the instance by
filling the missing effect function. Don't use any primitive outside the provided primitive list
corresponding to each instance, e.g., if there is no 'green_door' in the primitive list you must
not use 'green_door' for the effect function.

Advice = "Don't go to the door without the key"
Primitives = ['yellow_door', 'goal', 'pickup', 'yellow_key', 'toggle', 'go_to', 'carrying', 'at']

```
Effect main:
  if at(yellow_door) and not carrying(yellow_key):
    Reward -1
```

Advice = "Don't walk into closed doors. If you're tired, don't go forward."
Primitives = ['Agent', 'Wall', 'GoalTile', 'Lava', 'Key', 'Door', 'Box', 'Ball', 'left', 'right',
'forward', 'pickup', 'drop', 'toggle', 'done', 'pointing_right', 'pointing_down', 'pointing_left',
'pointing_up', 'go_to', 'step_towards', 'green_ball', 'green_box', 'purple_box', 'agent', 'pur-
ple_ball', 'at', 'reachable', 'carrying']

```
Effect main:
  if at(yellow_door) and yellow_door.is_closed and A == forward:
    Reward -1
    S' -> S
  elif tired() and A == forward:
    Reward -1
```

Advice = "Walking into balls is pointless. You will die if you walk into keys. Trying to open
a box when you aren't near it will do nothing."
Primitives = ['Agent', 'Wall', 'GoalTile', 'Lava', 'Key', 'Door', 'Box', 'Ball', 'left', 'right',
'forward', 'pickup', 'drop', 'toggle', 'done', 'pointing_right', 'pointing_down', 'pointing_left',
'pointing_up', 'go_to', 'step_towards', 'green_ball', 'green_box', 'purple_box', 'agent', 'pur-
ple_ball', 'at', 'reachable', 'carrying']

```
Effect main:
  if at(Ball) and A == forward:
    Reward 0
    S' -> S
  elif at(Key) and A == forward:
    Reward -1
    S' -> S*0
  elif at(Box) and A == toggle:
    Reward 0
    S' -> S
```

---

A.2  USER STUDY - TRANSLATED ADVICE

Table 3: Advice from the user study translated to RLang.

| Language Advice | RLang Translation |
|---|---|
| "Remember to toggle to open doors." | ```Policy main:```
```  if at(yellow_door) and not```
```  yellow_door.is_open:```
```    Execute toggle```
```  elif at(red_door) and not red_door.```
```  is_open:```
```    Execute toggle```
```  elif at(purple_door) and not```
```  purple_door.is_open:```
```    Execute toggle```
```  elif at(blue_door) and not```
```  blue_door.is_open:```
```    Execute toggle```
```  elif at(green_door) and not```
```  green_door.is_open:```
```    Execute toggle```
```  elif at(grey_door) and not```
```  grey_door.is_open:```
```    Execute toggle``` |
| "You don't need to carry keys to open the grey door." | ```Effect main:```
```  if at(grey_door) and carrying(```
```  red_key):```
```    S' -> S```
```    Reward -1```
```  elif at(grey_door) and```
```  carrying_something():```
```    S' -> S```
```    Reward -1``` |
| "Identify the room with the red key, move to that room by opening the door. Pick up the key. Identify the room with the red door, proceed there. Open the red door. Find the green square and go there to finish the game." | ```Plan main:```
```  Execute go_to(red_key)```
```  Execute pickup```
```  Execute go_to(red_door)```
```  Execute toggle```
```  Execute go_to(goal)``` |
| "Move to the grey door, open it and enter the room until you get to the red key, pick it up. Exit the room and move towards the red door, open it and get into that room. Move to the green block and enter it." | ```Plan main:```
```  Execute go_to(grey_door)```
```  Execute toggle```
```  Execute go_to(red_key)```
```  Execute pickup```
```  Execute go_to(grey_door)```
```  Execute toggle```
```  Execute go_to(red_door)```
```  Execute toggle```
```  Execute go_to(goal)``` |
| "Go to the grey door. open the grey door. go to the red key. pick up the red key. go to the red door. open the red door. go to the green square." | ```Plan main:```
```  Execute go_to(grey_door)```
```  Execute toggle```
```  Execute go_to(red_key)```
```  Execute pickup```
```  Execute go_to(red_door)```
```  Execute toggle```
```  Execute go_to(goal)``` |

Table 4: Advice from the user study translated to RLang (continued).

| Language Advice | RLang Translation |
|---|---|
| "Pick up the red key after opening the grey door. Then walk to the red door, open it, and go to the goal." | ```Plan main:    Execute go_to(grey_door)    Execute toggle    Execute go_to(red_key)    Execute pickup    Execute go_to(red_door)    Execute toggle    Execute go_to(goal)``` |
| "You cannot open the red door without a red key." | ```Effect main:    if at(red_door) and not carrying(    red_key):       S' -> S       Reward -1``` |
| "Walking towards the red door is not very useful if it is closed." | ```Effect main:    if at(red_door) and not(red_door.    is_open) and A == forward:       S' -> S       Reward -1``` |
| "Go down until the second door on the left and pick up the key. Then exit the room and go down until the next door on the left and use it to open the door and get to the green box." | ```Plan main:    Execute go_to(second_left_door)    Execute pickup    Execute go_to(exit)    Execute go_to(next_left_door)    Execute toggle    Execute go_to(green_box)``` |
| "Go to the room that has the red key, pick it up, and then go to the room with a red door. Enter the room, and go to the green goal object." | ```Plan main:    Execute go_to(red_key)    Execute pickup    Execute go_to(red_door)    Execute toggle    Execute go_to(goal)``` |

## A.3   MidMazeLava - Translated Advice

Advice: "Pick up the blue ball and drop it to your right. Then pick up the green key and unlock the green door. Then drop the key to your right. Some general advice: If you are carrying a key and its corresponding door is closed, open the door if you are at it, otherwise go to the door if you can reach it. Otherwise, drop any keys for doors you can't reach. If you can reach the goal, go to it. Walking into lava will kill you. If you're not at a door, toggling will do nothing. Trying to pick something up while you're carrying something is pointless. Walking into walls will do nothing."

```
Plan main:
  Execute go_to(blue_ball)
  Execute pickup
  Execute right
  Execute drop
  Execute go_to(green_key)
  Execute pickup
  Execute go_to(green_door)
  Execute toggle
  Execute right
  Execute drop

Policy main:
  if carrying(green_key) and not green_door.is_open:
    if at(green_door):
      Execute toggle
    elif reachable(green_door):
      Execute go_to(green_door)

  elif carrying(grey_key) and not grey_door.is_open:
    if at(grey_door):
      Execute toggle
    elif reachable(grey_door):
      Execute go_to(grey_door)

  elif reachable(goal):
    Execute go_to(goal)

  elif carrying(green_key) and not reachable(green_door):
    Execute drop

  elif carrying(grey_key) and not reachable(grey_door):
    Execute drop

Effect main:
  if at(Lava) and A == forward:
    S' -> S*0
    Reward -1
  if not at(Door) and A == toggle:
    S' -> S
    Reward 0
  if carrying_something() and A == pickup:
    S' -> S
    Reward 0
  if at(Wall) and A == forward:
    S' -> S
    Reward 0
```

## A.4 HARDMAZELIGHT - TRANSLATED ADVICE

Advice: "Go and pick up the green ball, and drop it on your left, and then go pick up the blue key, and go to the blue door and open it up and drop the key on your left, and then go pick up the green key, and go to the green door to open it and drop the key on your left, and then go pick up the purple ball and drop it on your right. Nothing will happen if you walk towards the wall, or try to open a purple door without the purple key if it is locked. The applies for the yellow door and key as well as the red door and key. If you can reach the grey door and it is closed but you have the key, open it if you are at it or otherwise go to it. The same applies to the purple door, yellow door, and red door. Lastly, if you find the goal is reachable just go to the goal directly."

```
Plan main:
    Execute go_to(green_ball)
    Execute pickup
    Execute left
    Execute drop
    Execute go_to(blue_key)
    Execute pickup
    Execute go_to(blue_door)
    Execute toggle
    Execute left
    Execute drop
    Execute go_to(green_key)
    Execute pickup
    Execute go_to(green_door)
    Execute toggle
    Execute right
    Execute drop
    Execute go_to(purple_ball)
    Execute pickup
    Execute right
    Execute drop

Effect main:
    if at(Wall) and A == forward:
        Reward 0
        S' -> S
    elif at(purple_door) and purple_door.is_locked and A == toggle and
     not carrying(purple_key):
        Reward 0
        S' -> S
    elif at(yellow_door) and yellow_door.is_locked and A == toggle and
     not carrying(yellow_key):
        Reward 0
        S' -> S
    elif at(red_door) and red_door.is_locked and A == toggle and not
    carrying(red_key):
        Reward 0
        S' -> S
```

```
Policy main:
  if reachable(grey_door) and carrying(grey_key) and grey_door.
  is_locked:
    if at(grey_door):
      Execute toggle
    else:
      Execute go_to(grey_door)

  elif reachable(purple_door) and carrying(purple_key) and
  purple_door.is_locked:
    if at(purple_door):
      Execute toggle
    else:
      Execute go_to(purple_door)

  elif reachable(yellow_door) and carrying(yellow_key) and
  yellow_door.is_locked:
    if at(yellow_door):
      Execute toggle
    else:
      Execute go_to(yellow_door)

  elif reachable(red_door) and carrying(red_key) and red_door.
  is_locked:
    if at(red_door):
      Execute toggle
    else:
      Execute go_to(red_door)

  elif reachable(goal):
    Execute go_to(goal)
```

# B  ADDITIONAL EXPERIMENT: GROUNDING COMMANDS TO RLANG PLANS

We compare our method to SayCan (Ahn et al., 2022), which uses the commonsense reasoning capacity of LLMs to satisfy a natural language request by generating a simple plan consisting of a series of pre-engineered high-level robot skills. Adopting the same

In this experiment we demonstrate that, in a simulated 3-dimensional physical environment, RLang can express the full range of natural language instructions necessary for a robot to complete various tasks. By grounding natural language instructions to RLang policies over this environment, we achieve performance on par with the results from the open-source tasks that the original SayCan paper evaluated on, showing that RLang can be easily substituted for the formal language that the SayCan authors developed for this specific task, allowing for generalization without sacrificing performance.

Similar to the SayCan work, we assume that we are given a grounding tuple $\langle \Pi, S, A, \rangle$, and a set of skills $\Pi$, where each skill $\pi \in \Pi$ performs an action with the robot arm to manipulate a block or a bowl. We evaluate on the 8 unique tasks made available in the open-source version of SayCan, running each task across 10 different randomly selected initial states, using both the native SayCan language and RLang as the DSL for grounding natural language instructions to robot behavior.

Each task configuration that the original SayCan agent completes, is also completed by the RLang agent. While their behavior on failure cases occasionally varied, these were generally caused by errors in the vision model's processing of shadows in the simulated environment. These generally caused the textual scene description fed into GPT-3 to include a block where a bowl should be, and occasionally incorrect color labels, which often provided the text-only planner with a nonsensical task that was impossible to complete. Similarly, in cases where multiple action orders could satisfy the request, the RLang and SayCan pipelines occasionally diverged in the order of actions. Nonetheless, neither language grounding pipeline completed a task configuration that the other one did not.

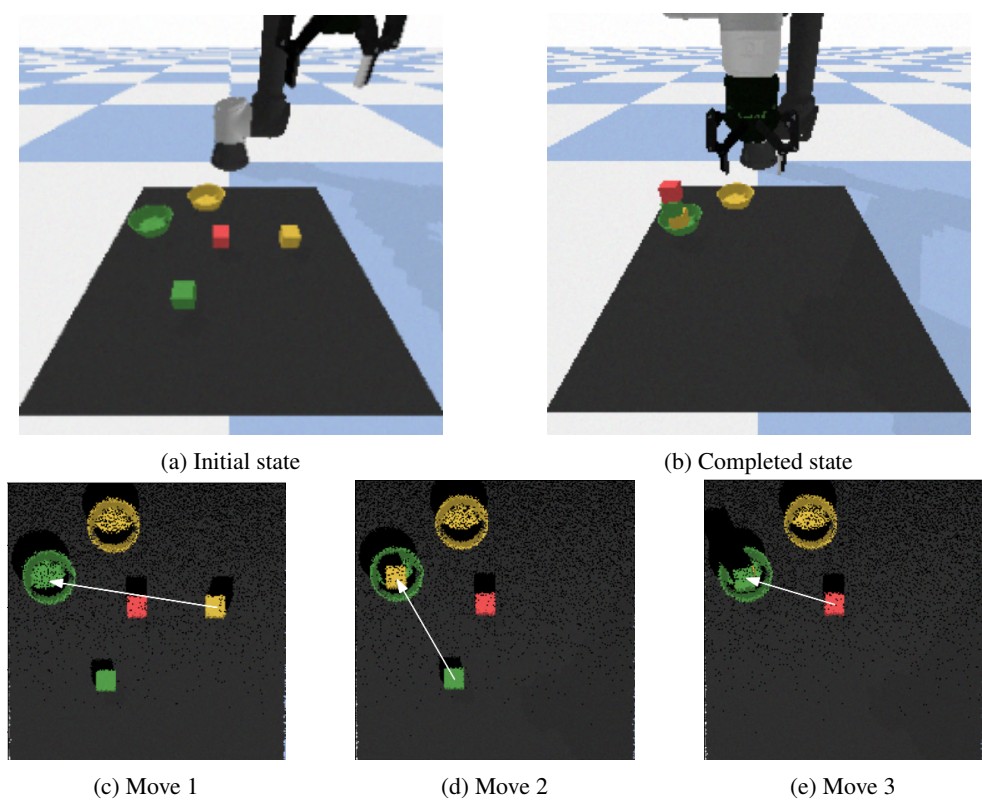

Figure 7: One configuration of the SayCan environment, and the action sequence to execute on the instruction: "put all the blocks in the green bowl."

Table 5: Success rates of SayCan and RLang-based instruction grounding rate on each task, out of 10 random initial states.

| Instruction | SayCan | RLang |
|---|---|---|
| put all the blocks in different corners. | 10 | 10 |
| move the block to the bowl. | 6 | 6 |
| put any blocks on their matched colored bowls. | 7 | 7 |
| put all the blocks in the green bowl. | 7 | 7 |
| stack all the blocks. | 8 | 8 |
| make the highest block stack. | 7 | 7 |
| put the block in all the corners. | 10 | 10 |
| clockwise, move the block through all the corners. | 10 | 10 |

