# OpenReview forum: "Informing Reinforcement Learning Agents by Grounding Natural Language to Markov Decision Processes"
_ICLR.cc/2024/Conference — ICLR 2024 Conference Withdrawn Submission_

### Official Review · Reviewer_42jA · 2023-10-26

**Soundness:** 2 fair
**Presentation:** 2 fair
**Contribution:** 2 fair
**Rating:** 3
**Confidence:** 3

**Summary:**

The paper addresses the challenge of efficiently utilising diverse form of natural language to improve RL performance. To do this, the paper proposes an approach to translate natural language into RLang (Rodrigueze-Sanchez et al. 2023), a formal language designed to express information about an MDP and its solution, and introduces RLang-Dyna-Q, a model-based framework which can utilise the translated advice.

**Strengths:**

- The studied problem of grounding natural language advice to help accelerate RL is interesting and important
- I like the idea of translating natural language advice to formal RLang and using it to construct a model of the transition dynamics and reward to help improve RL

**Weaknesses:**

- The assumption of deterministic MDP limits the broader application of this method
- At the start of section 4 Experiments: the authors mentioned the first hypothesis to be evaluated: RLang is an expressive grounding for natural language advice in the context of RL. It is unclear how this hypothesis is validated.
  - What is the definition of expressive? Is there a concrete metric defined to measure expressiveness and how does RLang perform?
  - What if an agent started in a different position of the maze? Are the language advice still valid?
  - In figure 5 (c), the advice is "pick up the blue ball and drop it to your right".  Here "to your right" is ambiguous, because it would depend on the direction in which the agent is facing, and how many grids to the right of the agent is another point of ambiguity. It is unclear to me how such ambiguity in natural language can be expressed accurately by RLang.

**Questions:**

In Table 3. "You don't need to carry keys to open the grey door" translated to a ```if at (grey_door) and carrying (red_key): S' -> S, Reward -1; elif at(grey_door) and carrying_something(): S' -> S, Reward -1```,  while in Table 4. "You cannot open the red door without a red key" translated to ```if at (red_door) and not carrying (red_key): S' -> S Reward -1```.

Is the reward of -1 meaningful or is it just a step penalty? Because the two cases seems quite different but the rewards are the same. And I'm curious why all rewards in the table are -1?

---

### Official Review · Reviewer_Eou7 · 2023-10-30

**Soundness:** 2 fair
**Presentation:** 2 fair
**Contribution:** 2 fair
**Rating:** 3
**Confidence:** 4

**Summary:**

This paper focuses on improving RL with the use of grounded formal language that can map to every component of a MDP. The authors argue that existing methods, which map natural language to individual elements of an MDP, have a limited scope. To address this, they introduce a method to translate general language advice to a grounded formal language. The authors propose RLang-Dyna-Q, where agents were grounded with various pieces of natural language advice, leading to improvements in performance. The evaluation shows that RLang can effectively translate natural language advice into discrete robotic actions without any loss in efficiency.

**Strengths:**

- This paper proposes a unique method, and the author claims that it can solve the limitations of previous methods.
- This paper evaluates RLang-Dyna-Q in two different environments, demonstrating improvement.

**Weaknesses:**

- The experiment is not solid enough. Lack of comparison with other state-of-the-art methods for mapping natural language to individual elements.
- Lack of analysis on the experimental results regarding the language advice used.
- RLang does not provide language advice for filter processing, which should have high quality requirements for the data used.

**Questions:**

- How does RLang-Dyna-Q compare with current state-of-the-art methods which map natural language to individual elements?
- While the method has shown to comprehend and learn from the advice given, how does it react to conflicting advice?
- Is the process of translating general language advice to RLang dependent on the quality of advice provided? If so, how does it filter useful information from irrelevant or incorrect advice?
- Could there be potential issues related to overfitting when the agent relies too heavily on the advice provided?

---

### Official Review · Reviewer_LPUN · 2023-10-31

**Soundness:** 2 fair
**Presentation:** 3 good
**Contribution:** 2 fair
**Rating:** 3
**Confidence:** 4

**Summary:**

This paper studies grounding the elements in RL using languages. The authors proposed a model-based reinforcement learning algorithm called RLang-Dyna-Q.

**Strengths:**

1. The paper's organization is clear and easy to follow.
2. The method is straightforward and easy to implement, which is good.

**Weaknesses:**

1. The novelty is limited. The idea of instantiating different components in an RL algorithm using LLMs is not novel. In fact, there are some papers that also used model-based approaches characterized by the LLM, such as [1-3]. The authors are encouraged to include these related works in their paper.
2. The method works in a zero-shot manner, i.e., the LLM generates the model, reward etc before execution with provided prompts. The algorithm does not fully leverage the reasoning and reflecting abilities in LLMs, e.g., by summarizing errors made in approximating transitions and rewards. The performance of the algorithm highly depends on the quality of the prompts.
3. The experiments are only conducted in very simple tabular environments, where the dynamics are pretty simple to infer. I have concerns regarding the effectiveness of the algorithm in more complex domains, where it would be necessary to incorporate reflection procedures.
4. "Language advice" does not sound like a formal term and should be explicitly defined in the RL context.

[1] Reasoning with Language Model is Planning with World Model. Hao et al.
[2] Reason for Future, Act for Now: A Principled Framework for Autonomous LLM Agents with Provable Sample Efficiency. Liu et al.
[3] LanguageMPC: Large Language Models as Decision Makers for Autonomous Driving. Sha et al.

**Questions:**

Why does the performance of the random policy decay as more episodes are taken?

---

### Official Review · Reviewer_qode · 2023-11-04

**Soundness:** 3 good
**Presentation:** 3 good
**Contribution:** 2 fair
**Rating:** 3
**Confidence:** 4

**Summary:**

The authors propose to use an LLM to automatically generate formal language (RLang) for a language conditioned MDP. They assume the environment is already suitable for RLang, e.g. the user must specify groundings between language and actions, language and perception ,etc. The LLM is able to generate advice in RLang that impacts many parts of the MDP, like rewards, transition functions, policies, etc. They evaluate their method, RLang-Dyna-Q in gridworld environments and show significant improvements over no-language RL.

**Strengths:**

The high level motivation of integrating language into all parts of the RL process is compelling.

The use of the LLM to output advice reduces a good amount of human effort, and makes sense for generalization.

I like some parts of the experiments, such as conducting a user study to test how well RLang can ground "in-the-wild" natural language advice, and the ablations that analyze which components (Q function, plans, policies) benefit the most from advice.

**Weaknesses:**

My most pressing concern about this paper, is its narrow experimentation in toy domains, and lack of  baselines. The authors choose Minigrid environments as an idealized setting for language-conditioned RL. They write:

>  Minigrid environments are an ideal setting for our experiments for three reasons: 1) they can be solved using tabular RL algorithms, which our informed, model- based RLang-Dyna-Q agent is based on; 2) there are clear and obvious referents of language in both the state and action spaces of these environments (e.g. keys, doors, and balls are represented neatly in a discrete state space and skills such as walking towards objects are easy to implement); 3) many objects are shared across environments enabling the reuse of a common RLang vocabulary for referencing these objects, which makes it easier for our translation pipeline to ground novel advice.

I believe a large amount of hard RL environments, particularly in robotics or 1st-person video game environments, break all of these assumptions, and make language-conditioned RL difficult. While asking the authors to solve all of these environments is a bit unfair, I would like to see the authors tackle some harder environments with these properties. Otherwise, I believe it will be hard to convince outsiders to use these language-conditioned RL.

Next, the authors do not have any baselines, only ablations. There are many methods that the authors mention in the related work that could serve as baselines. I also suspect that in these extremely simple Minigrid environments, an LLM can just directly solve it.

**Questions:**

Can the authors evaluate in harder tasks with more realistic assumptions? A good way to approach this is to select environments where each constraint is violated (the degree of violation can also be varied), and show how the method and other baselines degrade with respect to the violation.

I suspect that the ablations trends vary with the task. It would be interesting to conduct more ablations on different tasks.

Can the authors compare against actual language RL baselines? I also suspect a simple LLM-only baseline can do quite well in simple environments.